# FEL Pulse Duration Evolution along Undulators at FLASH

**Mahdi M. Bidhendi *** , **Ivette J. Bermudez Macias †** , **Rosen Ivanov †** , **Mikhail V. Yurkov and Stefan Düsterer ***

Deutsches Elektronen-Synchrotron DESY, Notkestr. 85, 22607 Hamburg, Germany;
ivette.bermudez@xfel.eu (I.J.B.M.); rosen.ivanov@xfel.eu (R.I.); mikhail.yurkov@desy.de (M.V.Y.)
* Correspondence: mahdi.mohammadi-bidhendi@desy.de (M.M.B.); stefan.duesterer@desy.de (S.D.)
† Current address: European XFEL GmbH, Holzkoppel 4, 22869 Schenefeld, Germany.

**Abstract:** Self-amplified spontaneous-emission (SASE) free-electron lasers (FELs) deliver ultrashort pulses with femtosecond durations. Due to the fluctuating nature of the radiation properties of SASE FELs, characterizing FEL pulses on a single-shot basis is necessary. Therefore, we use terahertz streaking to characterize the temporal properties of ultrashort extreme ultraviolet pulses from the free-electron laser in Hamburg (FLASH). In this study, pulse duration as well as pulse energy are measured in a wavelength range from 8 to 34 nm as functions of undulators contributing to the lasing process. The results are compared to one-dimensional and three-dimensional, time-dependent FEL simulations.

**Keywords:** free-electron lasers; temporal diagnostic; XUV pulses; SASE; THz streaking

## 1. Introduction

The electron bunches in single-pass, high-gain, free-electron lasers (FEL) propagate through the undulators just once producing the most intense extreme ultraviolet (XUV) and X-ray pulses. The amplification mechanism is the so-called self amplification of spontaneous emission (SASE) resulting in fundamental statistical fluctuations in the radiation properties. The properties of radiation pulse energy and radiation pulse duration have been studied theoretically and experimentally in the last decades for XUV and X-ray ranges [1–4]. It turns out that the pulse energy (i.e., the number of photons in one ultrashort pulse) grows gradually by many orders of magnitude in the amplification process, whereas the pulse duration first decreases in the exponential stage of amplification and then grows by about a factor of two when the amplification process enters the nonlinear regime.

The evolution of the pulse energy along the undulators, the so-called gain curve, has been measured frequently in the past [5–11]. However, to our knowledge, there has been only one study that investigates pulse duration in relation to the number of undulators contributing to the lasing [4]. In this study, however, pulse duration is only estimated by analyzing the spectral fluctuations for one X-ray wavelength. Here, we present a more comprehensive study where we measured the single-shot pulse duration as well as the pulse energy for six different wavelengths at the FLASH2 facility [11]. In addition to the average, we can analyze the single-shot data and provide experimental values for the shot-to-shot fluctuations. To obtain a complete picture of the process, the experimental data are compared with FEL simulations.

## 2. SASE FEL Amplification Process

The amplification process in SASE FELs develops from the shot noise in the electron beam, passes an exponential stage of amplification, and finally evolves in the nonlinear regime. Figure 1 shows the evolution of the radiation pulse energy and its fluctuations, as well as the evolution of the radiation pulse duration along the undulator. The results are obtained by the time-dependent FEL simulation code FAST using 1D and 3D FEL models [12]. To be specific, we consider the case of an electron beam with a longitudinal

Gaussian profile with an rms duration $\tau_{el}$. Fluctuations in the radiation pulse energy reach a maximum value at the end of the exponential gain regime and subsequently decrease in the nonlinear regime, while the radiation pulse energy continues to grow. In the high-gain exponential regime, the number of modes $M$ in the radiation pulse is defined as the inverse-squared deviation of the radiation pulse energy, $M = 1/\sigma_E^2$, where $\sigma_E^2 = \langle (E_r - \langle E_r \rangle)^2 \rangle / \langle E_r \rangle^2$ [2,13]. The saturation point corresponds to the maximum brilliance of the radiation [14,15]. At the same time, the fluctuations in radiation pulse energy decrease by a factor of three with respect to the maximum value. In the framework of the one-dimensional model, the saturation length and coherence time at saturation given by

$$z_{sat} \simeq \frac{\lambda_W}{4\pi\rho}\left(3 + \frac{\ln N_c}{\sqrt{3}}\right), \quad (\tau_c)_{max} \simeq \frac{1}{\rho\omega}\sqrt{\frac{\pi\ln N_c}{18}}, \tag{1}$$

are expressed in terms of the FEL parameter $\rho$ [16] and the number of cooperating electrons $N_c = I/(e\rho\omega)$ [2,13,17]. Here, $\omega$ is the radiation frequency, $I$ is the beam current, $-e$ is the charge of the electron, and $\lambda_W$ is the undulator period.

A practical estimate for the parameter $\rho$ comes from the observation that in the parameter range of SASE FELs operating in the VUV and X-ray wavelength ranges, the number of field gain lengths to reach saturation is about 10 [14]. Thus, the parameter $\rho$ and the coherence time $\tau_c$ relate to the saturation length by:

$$\rho \simeq \lambda_W/z_{sat}, \quad \tau_c \simeq \lambda z_{sat}/(2\sqrt{\pi}c\lambda_W). \tag{2}$$

For the number of modes $M \gtrsim 2$, the rms electron pulse length $\tau_{el}$ and the minimum radiation pulse length $\tau_{ph}^{min}$ given in full-width half-maximum (FWHM) at the end of the exponential gain regime are given by [3,18,19]:

$$\tau_{ph}^{min}(FWHM) \simeq \tau_{el} \simeq \frac{M\lambda}{5\rho} \simeq \frac{M\lambda z_{sat}}{5c\lambda_W}. \tag{3}$$

The minimum radiation pulse duration expressed in terms of coherence time (given in Equation (2)) is

$$\tau_{ph}^{min}(FWHM) \simeq 0.7 \times M \times \tau_c. \tag{4}$$

The radiation pulse duration is mainly defined by the length of the lasing fraction of the electron bunch with some corrections related to the slippage effect. In the beginning of the amplification process, the radiation pulse shape just repeats the longitudinal shape of the electron bunch. In the exponential high-gain regime, the power amplification (and beam bunching) is stronger for higher currents; thus, the radiation pulse duration is reduced as the electron bunch travels along the undulator. When the amplification approaches saturation (full bunching) in the central part of the bunch, the tails of the electron bunch begin to contribute more to the radiation power. Beam bunching continues to grow there and the radiation pulse duration starts to grow as well. The effect of the lasing tails gives the same relative radiation pulse lengthening, as is illustrated in Figure 1. The pulse length at the saturation point is about 1.4 times higher than the minimum pulse for the linear regime given by Equation (2), and it is increased further up to about a factor of two in the deep nonlinear regime. The second effect leading to pulse lengthening is the slippage of the radiation by one radiation wavelength per one undulator period. Evidently, the slippage effect is more pronounced for shorter pulses and longer wavelengths.

The case of "cold" (zero energy spread) and monoenergetic electron bunches has been analyzed in earlier papers in the framework of a 1D FEL model [2,3,19,20]. Using the normalized electron pulse duration $\bar{\tau}_{el} = \rho\omega\tau_{el}$ allows us to describe the simulation results in a universal way for $\bar{\tau}_{el} \gtrsim 2$. The simulation results normalized this way show almost identical behavior for different $\bar{\tau}_{el}$, as illustrated in Figure 1. Note that for $\bar{\tau}_{el} \gtrsim 2$, $\bar{\tau}$ is essentially equal to the number of radiation modes ($\bar{\tau}_{el} \sim M$) [3,19]. The one-dimensional model allows us to describe the physics of pulse length effects and temporal properties of

the radiation in an elegant way involving a minimum number of parameters. However, to describe a real experiment more quantitatively, a three-dimensional model which describes diffraction effects and the effects of betatron motion is needed. In addition, the details of the electron bunch structure (e.g., energy chirp) may be very important for the interpretation of the experimental results. Thus, in this paper we extend the analysis of short pulse effects using the results of three-dimensional, time-dependent simulations carried out with the FAST code. The bold curves in Figure 1 refer to the 3D simulations applying typical FLASH2 experimental conditions: electron energy of 1 GeV, rms energy spread of 0.2 MeV, and rms normalized emittance of 1.4 mm mrad. The lasing fraction of the electron bunch is approximated with a Gaussian distribution of 16 fs rms pulse duration and 1.5 kA peak current, resulting in a radiation wavelength of 13.5 nm. These parameters correspond to the value of $M \sim \bar{\tau}_{el} = 5.1$. The results of the simulations are presented with the same normalization procedure as for the 1D case. The first set of simulations refers to the case of a monoenergetic electron beam (bold black curves). We see that the 1D and 3D results for the monoenergetic case are rather similar, starting from the end of the high-gain exponential regime. The difference in the linear regime reflects the spatial mode competition effect which is absent in the 1D model. However, at the end of the high-gain linear regime, the fundamental $\mathrm{TEM}_{00}$ FEL mode is significantly larger as compared to other higher spatial modes, and we see a good agreement of the results of the 1D and 3D model.

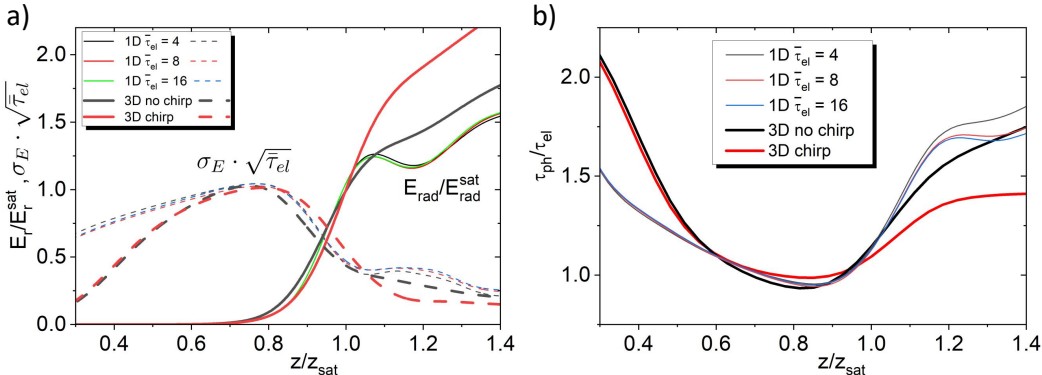

**Figure 1.** (**a**) Simulated average radiation pulse energy and fluctuations in the radiation pulse energy along the undulator length normalized to the saturation length. The fluctuations are scaled with $\sqrt{\bar{\tau}_{el}}$ to compensate for the pulse duration dependence as described below in the text. (**b**) Simulated evolution of the radiation pulse duration along the undulator length. Thin curves show the result of the 1D model for three different electron pulse lengths $M \sim \bar{\tau}_{el} = 4, 8$ and 16. The black bold curve shows the 3D model for the pulse length of $M \sim \bar{\tau}_{el} = 5$ without energy chirp, and the bold red curve shows the 3D model including energy chirp.

In the real accelerator, the electron beam is not monoenergetic. The long-pulse, low-current electron beam produced in the electron gun at FLASH is compressed in several stages by a large factor (up to about one hundred), and the peak current is increased correspondingly [21]. To achieve such a large compression, an energy chirp along the electron bunch is applied in the accelerating sections. This energy chirp leads to a bunch compression while the electron bunch moves through dedicated magnetic bunch compressors. An RF-induced energy chirp can be minimized such that it only slightly changes FEL properties with respect to a monoenergetic beam. Still, different kinds of wakefields and collective effects in the electron beam generate an energy chirp along the electron bunch. An example of such an energy chirp induced by the longitudinal space charge field (LSC) is shown in Figure 2 [21]. An important feature of an LSC wake is that the energy of electrons in the lasing fraction of the electron bunch is increased from the tail to the head of the electron bunch. Such a feature leads to a visible increase in the FEL efficiency in the nonlinear regime, which on the other hand leads to an increase in the FEL radiation bandwidth. During the experiments discussed in this paper, the beam formation system

was tuned such that the LSC produced a significant chirp of about 5 MeV peak-to-peak, resulting in an increase in the radiation spectrum bandwidth by about a factor of two with respect to the natural FEL bandwidth. The results of the simulation including the LSC chirped electron beam are shown in Figure 1 as a red bold line. While there is no difference between the chirped and monoenergetic (unchirped) cases in the linear regime, we see visible differences in the postsaturation regime. The pulse duration becomes shorter and the radiation pulse energy grows faster in the chirped case. The explanation of this phenomenon can be found in the positive energy chirp along the lasing fraction of the bunch (see Figure 2). It is well known that a linear energy chirp $\gamma/t$ along the bunch is equivalent to a linear undulator tapering such that [1]

$$\frac{1}{H_\mathrm{w}}\frac{dH_w}{dz} = -\frac{(1+K^2)^2}{2K^2}\frac{1}{\gamma^3}\frac{d\gamma}{cdt}, \tag{5}$$

where $H_w$ is the peak magnetic field and $K$ is the rms value of the undulator parameter. We see that the positive energy chirp is equivalent to an undulator tapering with decreasing field strength along the undulator. It is well known that the application of the undulator tapering with the decreasing field allows one to preserve the synchronism between electrons and the electromagnetic wave, thus increasing the FEL efficiency [13]. With an appropriate optimization of the energy chirp, one can realize conditions which are equivalent to an optimum undulator tapering, when a significant fraction of particles is trapped in the effective ponderomotive potential. The trapped electrons interact stronger with the radiation, and the radiation power grows along the undulator length, while the radiation pulse duration remains nearly constant and the slippage effect is suppressed as well. Of course, an increase in the energy chirp results also in an increase in the radiation bandwidth, as we see from the experimental data.

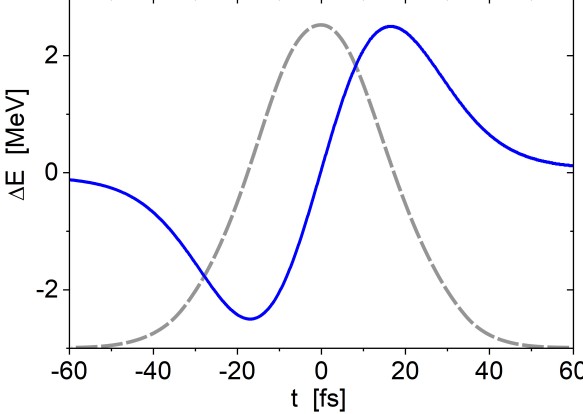

**Figure 2.** Energy chirp along electron bunch induced by the longitudinal space charge field (blue curve) as it was used for the chirped 3D simulation. The grey dashed curve shows the longitudinal profile of the electron bunch.

In the following sections, we compare experimental results with 3D simulations performed for a chirped electron beam. The energy chirp along the lasing fraction of the bunch was set to 150 keV/fs (Figure 2), leading to a spectral width of about 1% for the XUV pulse, in agreement with several spectral measurements performed at FLASH [5,22–24]. The chirp in the electron bunch leads to a chirp of 30 meV/fs in the 13.5 nm XUV pulse, or, expressed as second-order dispersion, it yields a value of 25 fs$^2$, which is comparable to the measurements described in Ref. [25], with a second-order dispersion of 50 fs$^2$, where an exceptionally large bandwidth (i.e., chirp) was requested.

### 3. FEL Measurements

In order to measure the pulse duration of the XUV FLASH pulses, we used the terahertz streaking technique [26–28]. In short, this method is based on the photoionization of noble gas atoms in the presence of a strong terahertz (THz) field. If the generated electron wave packet is shorter than half the streaking field period, the temporal structure of the wave packet will be mapped onto the kinetic energy distribution of the emitted electrons and can thus be used to determine the XUV pulse duration. The measurements were performed at the dedicated photon diagnostic beamline FL21 in the FLASH2 branch, which is equipped with a permanently installed THz streaking setup [29]. The setup consisted of the interaction chamber containing a time-of-flight spectrometer, a dedicated laser system delivering about 1 ps long pulses at 1030 nm with pulse energy of 3.5 mJ at a repetition rate 10 Hz, and a THz generation setup based on optical rectification using a nonlinear crystal LiNbO$_3$) (for details, see, e.g., [30,31]).

In Ref. [31], it was discussed that there is an optimum number of photoelectrons created in the ionization process by the FEL pulse. One needs sufficiently many electrons to record a single-shot photoelectron spectrum while limiting the number of electrons so that the resulting space charge effects are negligible. For our setup, this leads to an optimum XUV pulse energy in the range of several hundred nJ to few μJ, depending on the wavelength for neon as the target gas ($P_{neon} \sim 5 \times 10^{-7}$ mbar). To ensure the same experimental conditions for all measurements, the streaking setup was left unchanged during the gain curve measurements, while the transmission of a variable XUV attenuator [32] was adapted such that the average number of created photoelectrons was constant for each measured setting. The FEL was operating in the burst mode delivering three pulses (with 1 μs spacing), while the first pulse was streaked by the terahertz field and the second pulse was used as the reference photoelectron spectrum. The actual pulse duration was derived according to the methods described in [31].

FLASH2 is equipped with 12 variable-gap undulator segments, each of 2.5 m length and 31.4 mm period [11]. The pulse duration was measured with the streaking setup while varying the number of undulators contributing to the lasing. The shown data result from 3 different measurement campaigns. In the first one, FLASH2 was set to 3 different wavelengths: 8, 12, and 16 nm with a constant electron bunch energy of 1.00 GeV and an electron bunch charge of 0.19 nC, leading to XUV pulse durations on the order of 100 fs FWHM. For the second campaign, the FLASH2 electron bunch energy was 875 MeV with a bunch charge of 0.2 nC, and the wavelength was set to 10 and 20 nm, leading to XUV pulse durations ranging from 50 to 160 fs FWHM. Besides these two campaigns, the third campaign was performed with a significantly lower electron bunch energy of 434 MeV and an electron bunch charge of 0.2 nC, leading to a wavelength of 34 nm. The experimental results are summarized in Table 1.

**Table 1.** Main parameters derived from the measurements shown in Figure 3. The values were deduced from scaling the experimental data to the 3D simulation, with $z_{sat}$ as saturation length in "number of undulators" with a maximum number of undulators of 12 at FLASH2, $\tau_{ph}^{min}(FWHM)$ as minimum pulse duration, and $E_{sat}$ as the saturation pulse energy. In order to compare the experimental data to the 1D simulation, the normalized electron pulse duration $\bar{\tau}_{el}$ (equivalent to the number of modes M) was determined for all measured wavelengths using Equation (4), and the coherence time at saturation $\tau_c^{sat}$ (FWHM) was taken from Refs. [19,33].

| Wavelength | Electron Bunch Energy | $z_{sat}$ | $\tau_{ph}^{min}(FWHM)$ | $E_{sat}$ | $\tau_c^{sat}(FWHM)$ | $\bar{\tau}_{el}|M$ |
|---|---|---|---|---|---|---|
| 8 nm | 1008 MeV | 9.4 | 75 fs | 60 μJ | 7fs | 22 |
| 10 nm | 875 MeV | 8.9 | 50 fs | 39 μJ | 8 fs | 13 |
| 12 nm | 1008 MeV | 7.7 | 88 fs | 88 μJ | 9 fs | 16 |
| 16 nm | 1008 MeV | 7.0 | 105 fs | 130 μJ | 12 fs | 16 |
| 20 nm | 875 MeV | 6.1 | 95 fs | 50 μJ | 15 fs | 13 |
| 34 nm | 434 MeV | 8.3 | 77 fs | 26 μJ | 20 fs | 8 |

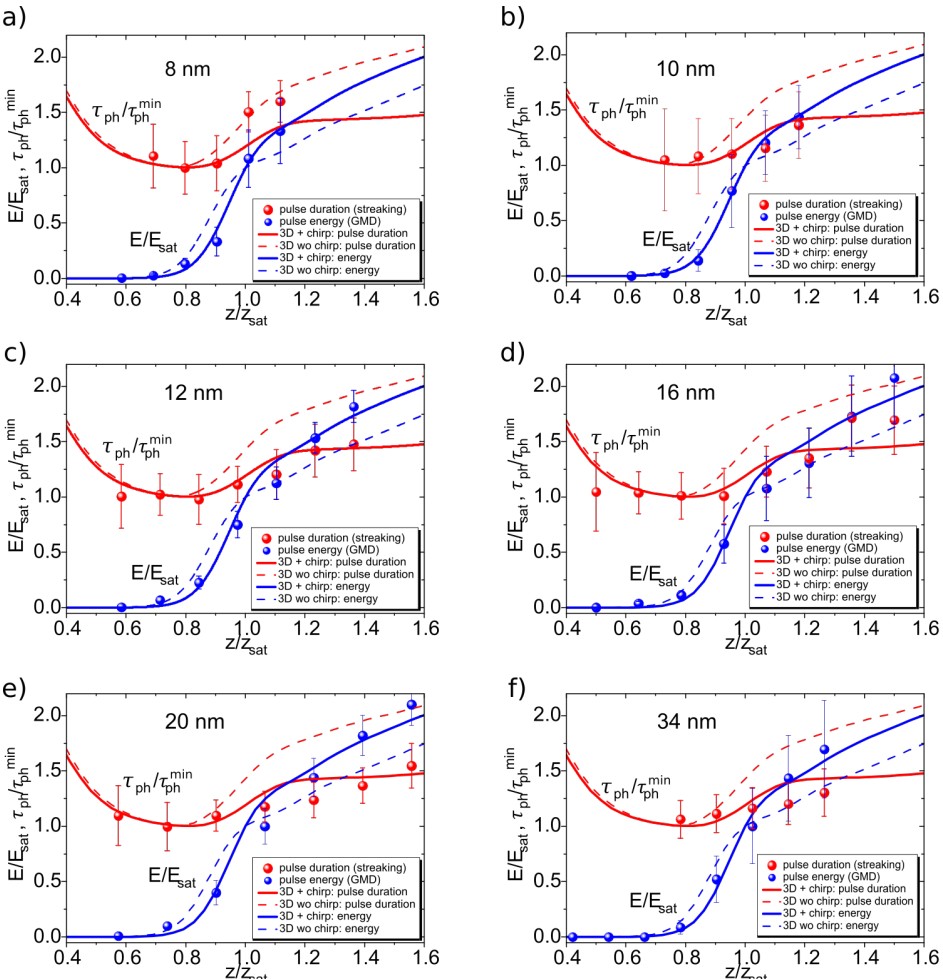

**Figure 3.** Evolution of the pulse duration (red circles) and pulse energy (blue circles) along the undulators. The actual undulator length $z$ is normalized by the saturation length $z_{sat}$. The pulse energy is normalized to the saturation energy and the pulse duration to the minimum pulse duration (Equation (4)). The experimental results (shown as points) represent the average over several thousand single pulse measurements and were recorded for FEL wavelengths of 8, 10, 12, 16, 20, and 34 nm and agree very well with the chirped 3D simulations (shown as red and blue solid lines). The 3D simulations without energy chirp (shown as red and blue dashed lines) are showing less agreement with the experimental data. The "error bars" denote not the experimental uncertainty but rather the width of the measured distribution induced by SASE and technical fluctuations to indicate by how much the FEL pulse energy and duration fluctuate during measurement (see also Figure 4).

For all measurements except at 34 nm, the undulators were not tapered (all undulators contributing to the lasing had the same gap settings, i.e., the same K-value), in order to compare the experimental values with the simulations. For each wavelength, the pulse duration was first measured when all 12 undulators were closed (thus, all undulators contributed to the lasing). Afterward, starting with the undulators closest to the experiments, one pair of undulators at a time was opened until no measurable XUV pulse energy could be detected. Since only the downstream undulators were opened, the trajectory in the first (lasing) undulators was kept constant. The resulting shift of the source point of the XUV radiation enlarged the XUV focal spot in the streaking interaction region slightly; however, it was still sufficiently smaller (<300 µm FWHM) as compared to the THz focal spot (~1 mm FWHM) and did not lead to significant changes for the THz streaking measurements. The energy of the XUV pulses was simultaneously measured with an absolutely calibrated pulse energy detector (GMD) [34] provided by the FLASH photon diagnostic.

## 4. Discussion

We analyzed the average pulse energies and average pulse durations as well as their respective fluctuations as functions of the number of undulators contributing to the lasing process for six different wavelengths. As a basis, we used the 3D simulation based on the chirped electron bunch. In the first step, the number of undulators was scaled to the undulator coordinate $z$ such that the onset of measurable SASE ($>0.5\ \mu J$) coincided with the $z$ range in which the energy gain became visible in the simulated energy gain curve ($z/z_{sat}\sim0.7$). This determined the undulator axis scaling and already fixed the saturation length $z_{sat}$. It has to be noted that we took into account that the first two undulator segments at FLASH2 hardly contributed to the lasing due to a slight misalignment, and thus for the experimental data we had to count the first two undulators as 0.5 undulators (i.e., subtracting 1.5 from the actual number of closed undulators). The simulation results were not affected by this technical issue. In the second step, the pulse energy was scaled such that the pulse energy at saturation length $z = z_{sat}$ was defined as saturation pulse energy $E_{sat}$ and all energies were normalized by the saturation energy. A similar normalization was applied for the pulse duration. Here, the measured values were scaled to the minimum pulse duration $\tau_{min}$ at $z/z_{sat}\sim0.8$. It is important to note that the experimental data have been only rescaled and not fitted to the simulation result.

Figure 3 shows the experimentally measured pulse duration (red spheres) as well as the 3D simulation including chirp (red line) and the 3D simulation without chirp (red dashed lines). The 3D model without chirp predicts a considerable pulse lengthening when the amplification process enters the nonlinear regime (similar to the 1D models shown in Figure 1), while the chirped simulation shows clearly less pulse lengthening. The experimental pulse duration data also do not indicate a strong lengthening of the XUV pulses in saturation and thus show a good agreement with the 3D simulation including chirp. For the pulse energy, we measured a steady increase in saturation and beyond. This behavior is again well represented by the 3D model including chirp, while the 3D model without chirp predicts less energy increase after the saturation point.

The pulse energies as well as the pulse durations were stochastically fluctuating due to the SASE process and fluctuations induced by the acceleration process as well as measurement uncertainties in the actual THz streaking measurement [19], leading to a broad distribution. Example histograms of the fluctuations measured for a specific setting (fixed wavelength and number of undulators) are shown in Figure 4b,d. The rms values of the fluctuations are also used as "error bars" in Figure 3, not denoting the measurement uncertainty but showing the range of the measured values. The actual experimental uncertainty for single-shot measurements was determined to be about ±20% for pulse duration measurements [31] and about ±10% for pulse energy measurements [19]; thus, only a small fraction of the shown "error bars" of the averaged values resulted from measurement uncertainties.

In Figure 4a,c, the normalized fluctuations are presented as functions of $z$. The fluctuations are not constant along the amplification process and for fewer closed undulators the fluctuations are much larger. Both the 1D and 3D simulations show that for pulse duration and pulse energy the relative fluctuations are largest in the linear regime and are decreasing strongly in the range of $z/z_{sat}$ 0.8 to 1 and only decrease slowly after saturation is reached.

In contrast to the simulated values of the pulse energy and the pulse duration (Figure 1) which are essentially pulse-length-independent, the respective fluctuations are indeed pulse-duration-dependent. We know from Ref. [19] that the fluctuations in pulse duration and pulse energy are inversely proportional to the square root of the number of modes, as is also indicated by the scaling in Figure 1a. Therefore, we expect less fluctuations for longer pulses (i.e., larger numbers of modes), as can be seen in Figure 4a,c. The 1D simulation was calculated for three different mode numbers, while the 3D simulation was only conducted for 5.1 modes, due to the larger computational load. Indeed, the 3D simulation results lay between the 1D simulation for four and eight numbers of modes for the pulse duration fluctuations and in the exponential gain regime for the energy fluctuations. However, it

is important to note that the 3D simulation predicts significantly less energy fluctuations as the 1D simulations in the saturation regime, which we could not verify experimentally due to technical reasons. Since the measured XUV pulse durations were in the range of $\bar{\tau}_{el}$ = 15–20, as shown in Table 1, they have to be compared mainly to the 1D simulation with $\bar{\tau}_{el} = 16$. The experimental values show essentially the same trend as the simulation calculated with, however, significantly higher fluctuations than the simulation predicts for the inherent SASE fluctuations. This observation was already described in [19] and can be attributed to additional fluctuations, including the measurement uncertainties, fluctuations in the energy gain, and compression of the electron bunches (due to acceleration field phase instability).

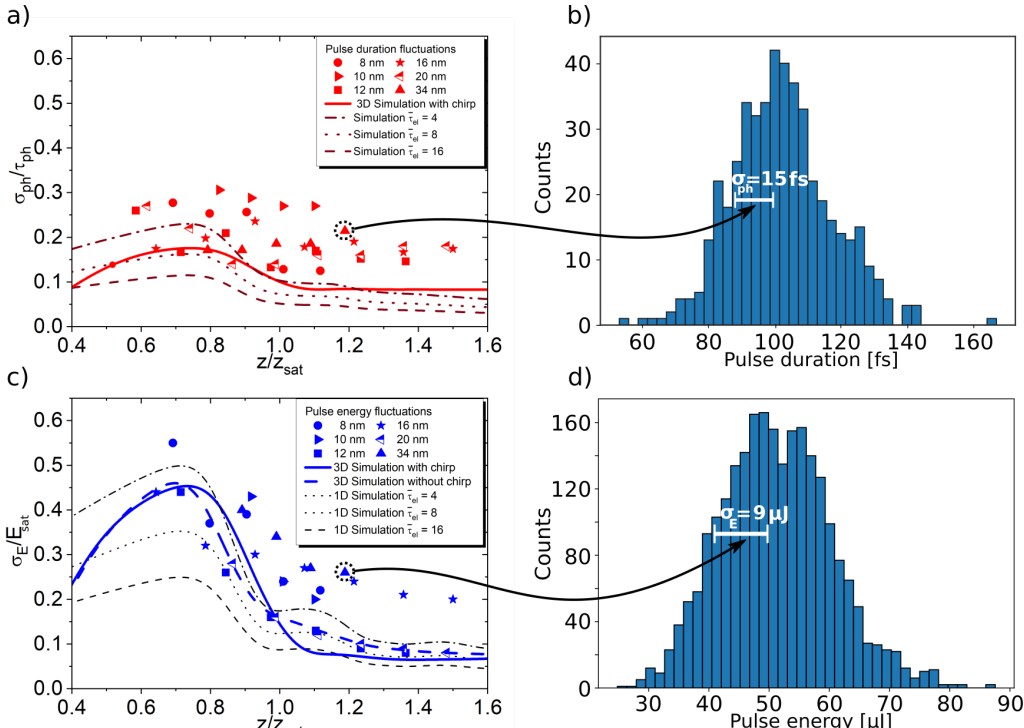

**Figure 4.** Shown are the fluctuations as functions of the undulator coordinate $z$ for (**a**) the pulse duration and (**c**) the pulse energy. For each setting, several thousand FEL pulses were measured, leading to a broad, Gaussian distribution as shown in (**b**) the pulse duration and (**d**) the pulse energy. The example histograms in (**b**,**d**) are corresponding to the data point shown with the dashed circle. The values for the fluctuations shown are the rms values of the measured pulse energies and pulse durations which are equivalent to the $1/e$ width of the histogram (for the nearly Gaussian distribution). In addition, the theoretically determined values for the fluctuations (only taking the SASE-induced effects into account) are plotted for the chirped and nonchirped 3D simulation and 1D simulation for the cases of $\bar{\tau}_{el} = 4$, 8, and 16. The fact that the measured fluctuations are larger than the theoretically expected values (only for SASE) can be attributed to measurement uncertainty and additional technical fluctuations in the accelerator, as was discussed in Ref. [19].

An interesting property for experiments is the radiation power (pulse energy divided by the pulse duration). In Figure 5a, we compare how the power scales with $z$. It turns out that the achievable power of the FEL pulses increases for all experimental settings continuously along the undulators, even in saturation without a hint for a local maximum. The chirped 3D simulation predicts the same behavior and shows a very good agreement with the experimental data. The nonchirped 3D simulation exhibits a similar trend in the linear regime and a small local maximum at the saturation point, which is not reproduced by the experimental data. In contrast, in the 1D simulation, the power reaches a distinct maximum saturation and grows again only for deeply saturated operation. By comparing

the results from the simulation methods, the chirped 3D simulation shows a much better agreement with the experimental data and should be used as a basis for the future prediction of FLASH radiation parameters.

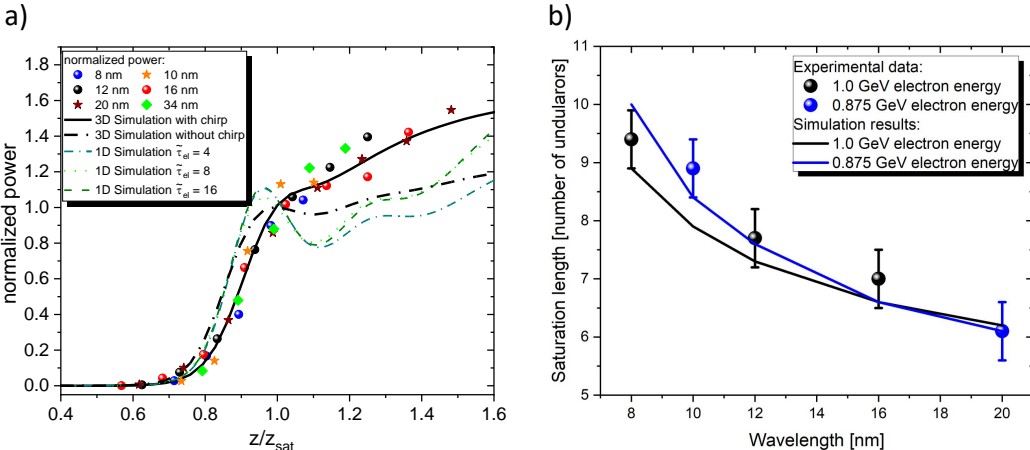

**Figure 5.** (**a**) Development of the power (pulse energy divided by pulse duration) along the undulators for different pulse durations. The chirped 3D model agrees significantly better with the experimental data as compared to the nonchirped 3D model, while the oscillatory behavior predicted by the 1D model is not reproduced at all. (**b**) The measured as well as the simulated (3D model with chirp) saturation length is plotted as functions of the FEL wavelength for two different FEL setups. Error bars are estimated as ±0.5 undulators from the analysis. The electron bunch energy for the 34 nm measurement was much smaller than for the other measurements; thus, it is not included in the plot.

Looking at Figure 5, one sees a continuous power increase along the undulator length. However, as pointed out in Refs. [14,15,35], considering the spatial and spectral properties of the FEL beam, it turns out that the maximum brilliance of the radiation is rather achieved in the very beginning of the nonlinear regime. Thus, for experiments that need the highest possible photon densities on the target, it may be beneficial to work close to the saturation point at $z/z_{sat} \sim 1$. Since neither spectral nor spatial measurements have been performed in this study, we cannot test this statement experimentally.

Comparing the FEL parameters (see Table 1) from the data shown in Figure 3, we observed that for the same FEL settings (8 nm, 12 nm, 16 nm and 10 nm, and 20 nm, respectively), the achievable saturation energy $E_{sat}$ increases with longer wavelength [11,36]. The minimal pulse duration ($\tau_{min}(FWHM)$) also increases slightly while the needed number of undulators to achieve saturation ($z_{sat}$) decreases. Plotting the saturation length as a function of the FEL wavelength together with the results from the chirped 3D simulation (Figure 5b) shows a good agreement. This again points out that the chirped 3D simulation can be used to reliably predict FLASH2 radiation parameters despite the different experimental settings (different electron energies, undulator gaps, and tuning).

## 5. Conclusions

We investigated FEL pulse duration and pulse energy and their fluctuations as functions of undulator length for six different FEL wavelengths. We compared the simulation results from chirped and nonchirped 3D FEL simulations with 1D FEL simulations. The chirp strengths were chosen such that the typical spectral widths (~1%) observed at FLASH were matched. Comparing the experimental results measured for different FEL setups with various simulation models, we found a compelling agreement with the chirped 3D model even if the FEL parameters deviated significantly between measurement and simulation. This indicates that the result does not depend strongly on the particular FEL setting. The features predicted by the 1D model and the nonchirped 3D model could not be reproduced.

In contrast, the chirped 3D simulation results describe the behavior of the pulse energy and pulse duration rather accurately for FLASH and can be well used as input for future experiments or comparison to measurements.

**Author Contributions:** Measurements, M.M.B., S.D., R.I. and I.J.B.M.; Simulation, M.V.Y.; writing—original draft preparation, M.M.B. and S.D.; writing—review and editing, M.M.B., S.D., R.I., I.J.B.M. and M.V.Y. All authors have read and agreed to the published version of the manuscript.

**Funding:** This research was funded by Deutsche Forschungsgemeinschaft (DFG, German Research Foundation) grant number 491245950.

**Institutional Review Board Statement:** Not applicable.

**Informed Consent Statement:** Not applicable.

**Data Availability Statement:** The data presented in this study are available on request from the corresponding author.

**Acknowledgments:** We want to acknowledge Franz Kärtner, Mikhail Pergament, Anne-Laure Calendron, Joachim Meier, Martin Kellert, and Simon Reuter for providing and maintaining our regenerative amplifier. We want to thank the DESY Synchronization group (in particular Sebastian Schulz) for providing an excellent synchronization of our laser. We would like to acknowledge Ulrike for useful discussions and suggestions for improving this article. We also thank the FLASH operators, in particular Juliane Rönsch-Schulenburg, for helpful discussions and for fulfilling our special wishes during our beamtimes.

**Conflicts of Interest:** The authors declare no conflict of interest.

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
