# Peer review of "FEL Pulse Duration Evolution along Undulators at FLASH"

_applsci, doi:10.3390/app12147048_

Round 1
Reviewer 1 Report
Interesting work that deserve to be published
Author Response
Dear reviewer,
we appreciate your positive feedback on our manuscript.
Reviewer 2 Report
This study describes the evolution of FEL pulse duration along the undulators in the high gain x-ray free-electron laser. The evolution of both pulse duration and pulse energy was measured for the first time. And the authors tried to compare the experimental result with the FEL code simulation and found that the measurement agrees with the chirped 3D model. Considered its first achievement, it deserves publication soon. The measurement is based on the well-proven THz streaking method that the co-authors published before. Thus the comparison with FEL simulation makes sense and shows that the energy chirp in the bunch plays a critical role in the FEL applications. It explains why amplification still takes place after saturation.
But the explanation about the post saturation amplification seems to be not enough (A short description with Fig. 2). More explanation or simulation is necessary to make this manuscript more readable.
The first two undulator segments at FLASH2 hardly contribute to the lasing due to the slight misalignment and thus the first 2 undulators were counted as 0.5 undulators (i.e. subtracting 1.5 from the actual number of closed undulators). This is not an appropriate way because too much uncertainty is involved in the simulation. It is not clear how to deal with those undulators in the simulation. Actually, the reasonable way is to omit those undulators in the measurement and simulation.
The following minor errors need to be corrected.
At line 30, “with” is missing. (compared with FEL simulations)
At line 52, the comma needs to be removed.
At line 265, I recommend rewriting it as “We investigated the FEL pulse duration and pulse energy and their fluctuations as a function of the undulator length for six different FEL wavelengths.”
Author Response
Dear reviewer,
we appreciate the time and effort that you dedicated to providing feedback on our manuscript and are grateful for your insightful comments on our paper. Please see below, in blue, the point-by-point response to your comments:
Point 1: The first two undulator segments at FLASH2 hardly contribute to the lasing due to the slight misalignment and thus the first 2 undulators were counted as 0.5 undulators (i.e. subtracting 1.5 from the actual number of closed undulators). This is not an appropriate way because too much uncertainty is involved in the simulation. It is not clear how to deal with those undulators in the simulation. Actually, the reasonable way is to omit those undulators in the measurement and simulation.
Response 1: We know that the approach to empirically subtract 1.5 undulators from the actually closed undulators is not optimal. During the measurements we were not aware of this problem which was discovered after the beamtimes. Different measurements done by the FEL operators later on showed that on average the first 2 undulators only contribute as about half an undulator. If we would have been aware of this, we had opened the first 2 undulators completely for our measurements. However, since beamtime at an FEL is rather difficult to get, there is no way to just repeat the experiments and we have to cope with the uncertainty which we attribute for in the error bars of the experimentally determined gain lengths (Fig 5b) with an error of +- 0.5 Undulators. This technical problem however has nothing to do with the simulations. The simulations represent the data for a well aligned FEL which we also should have now at FLASH after some modifications in the FL2 accelerator tunnel. To compare the experimental data correctly to the simulations we only had to modify the number of undulators that we used in the experiment - no modification was done to the simulation data.
we changed the text accordingly from:
It has to be noted that we took into account that the first two undulator segments at FLASH2 hardly contribute to the lasing due to the slight misalignment and thus we counted the first 2 undulators as 0.5 undulators (i.e. subtracting 1.5 from the actual number of closed undulators).
to:
It has to be noted that we took into account that the first two undulator segments at FLASH2 hardly contribute to the lasing due to a slight misalignment and thus for the experimental data we had to count the first 2 undulators as 0.5 undulators (i.e. subtracting 1.5 from the actual number of closed undulators). The simulation results are not affected by this technical issue.
The minor errors have been also corrected and can be seen in the revised manuscript.
Reviewer 3 Report
The paper by Bidhendi et al. describes experimental results for photon pulse characterization at the FLASH2 FEL and compare them with simulations.
The results are clearly presented and strongly support the conclusions. I therefore support this paper for publication on "Applied Sciences".
I have just a couple of typos to highlight and a request of additional information about a sentence.
Line 30
compared FEL simulations --> compared with FEL simulations
Line 226
As the number of modes used for the 3D simulation is 5.1, therefore--> The number of modes used for the 3D simulation is 5.1, therefore
Line 232
This observation was already described in [20] and can be attributed to additional fluctuations in the accelerator and 234
measurement uncertainties besides the SASE inherent fluctuations.
It'd be beneficial for the reader to shortly recall what these fluctuations are
Author Response
Dear reviewer,
we appreciate the time and effort that you dedicated to providing feedback on our manuscript and are grateful for your insightful comments on our paper. The minor errors and typos have been corrected and can be seen in the revised manuscript.
Please see below, in blue, the point-by-point response to your comments:
Point 1:
“This observation was already described in [20] and can be attributed to additional fluctuations in the accelerator and measurement uncertainties besides the SASE inherent fluctuations”.
It'd be beneficial for the reader to shortly recall what these fluctuations are
Response 1:
We changed the text accordingly from:
The experimental values show essentially the same trend as the 1D simulation with τel = 16, with however significantly higher fluctuations as the simulation predicts. This observation was already described in [20] and can be attributed to additional fluctuations in the accelerator and measurement uncertainties besides the SASE inherent fluctuations.
to:
The experimental values show essentially the same trend as the simulation calculated, with however significantly higher fluctuations as the simulation predicts for the inherent SASE fluctuations. This observation was already described in [20] and can be attributed to additional fluctuations including the measurement uncertainties, fluctuations in the energy gain and compression of the electron bunches (due to the acceleration field phase instability).
Reviewer 4 Report
In this paper, the authors measured the pulse energy and duration of FLASH2 with six wavelengths (8, 10, 12, 16, 20, 36 nm) conditions and evaluated them with various simulation models of 1D, non-chirped 3D, and chirped 3D. As a result, they summarized that the chirped 3D model could describe the pulse behavior, especially, the pulse energy and duration inFLASH2. I have heard only a few publications reports about the investigation of pule energy and duration of FEL conditions with several wavelength generations as far as I know. Therefore, this paper is worth publishing the Applied Science. The authors should consider the following point before the publication.
1, In the 4th paragraph of the 6th page, I do not understand the relation of the model numbers, the simulated values (pulse energy and duration), and the simulation model. According to the experiment, the model numbers were shown in the range of 15 - 20 in Fig. 1. However, the authors mentioned 5.1, 4, and 8 modes in the 3rd line. They should clarify how many mode numbers they used for the simulation and why they calculated and compared the pulse evaluation (Fig. 4, 5) with different mode numbers (4, 8, 16) in the simulation. And they also add more information about the relation between the model numbers and the pulse energy fluctuations and duration.
2, Although, "FWHM" is a well-known word in general, the authors would be polite to spell out it
Author Response
Dear reviewer,
we appreciate the time and effort that you dedicated to providing feedback on our manuscript and are grateful for your insightful comments on our paper. Please see below, in blue, the point-by-point response to your comments:
Point 1:
In the 4th paragraph of the 6th page, I do not understand the relation of the model numbers, the simulated values (pulse energy and duration), and the simulation model. According to the experiment, the model numbers were shown in the range of 15 - 20 in Fig. 1. However, the authors mentioned 5.1, 4, and 8 modes in the 3rd line. They should clarify how many mode numbers they used for the simulation and why they calculated and compared the pulse evaluation (Fig. 4, 5) with different mode numbers (4, 8, 16) in the simulation. And they also add more information about the relation between the model numbers and the pulse energy fluctuations and duration.
Response 1:
We changed the text accordingly from:
To compare the fluctuations predicted from the 1D and 3D model, the number of modes should be in the same range. As the number of modes used for the 3D simulation is 5.1, therefore we can compare it to the 1D simulation with 4 and 8 number of modes (τel = 4 and τel = 8). The result shows that the 3D simulation lays between the 1D simulation with 4 and 8 number of modes. Most measured XUV pulse durations were significantly longer (τel in the range of 15-20 as shown in table 1) and thus can be compared to the 1D simulation with τel = 16
to:
The 1D simulation was calculated for 3 different mode numbers, while the 3D simulation was only conducted for 5.1 number of modes, due to the larger computational load. Indeed, the 3D simulation results lay between the 1D simulation for 4 and 8 number of modes for the pulse duration fluctuations and in the exponential gain regime for the energy fluctuations. But it is to note that the 3D simulation predicts significantly less energy fluctuations as the 1D simulations in the saturation regime, which we could not verify experimentally due to technical reasons. Since the measured XUV pulse durations were in the range of τel = 15 - 20 as shown in table 1, they have to be compared mainly to the 1D simulation with τel = 16.
Point 2:
Although, "FWHM" is a well-known word in general, the authors would be polite to spell out it
Response 2:
The full form is added to the text
Reviewer 5 Report
The pulse duration (by the terahertz streaking technique) and energy (by GMD) vs undulator length for six wavelengths have been measured, and compared with simulations with 3D and 1D FEL models at FLASH. This is a more comprehensive work comparing with ref.4, which concludes the chirped 3D simulation results describe the behavior of the pulse energy and pulse duration rather accurately for FLASH and can be well used as input for future experiments or comparison to measurements. It sounds and can be published with few minor modifications,
1, It would be good if the authors can briefly explain the difference between 1D and 3D FEL models.
2, For the Section “3. FEL measurements”, a sample THz streaking spectrum/photoelectron spectrum at one FEL wavelength would be suggested to show how to deduce the FEL pulse duration, and some information about target gas (e.g., density), FEL repetition rate should be included.
3, The style of Figure 5 b) should keep the same as others, e.g., sticks and legend box shadow.
Author Response
Dear reviewer,
we appreciate the time and effort that you dedicated to providing feedback on our manuscript and are grateful for your insightful comments on our paper. The point-by-point response to your comments is attached.
